

# New insights into the role of GSK-3β in the brain: from neurodegenerative disease to tumorigenesis

Shenjin Lai[1,*], Peng Wang[2,*], Jingru Gong[1] and Shuaishuai Zhang[2,3]

[1] Department of Pharmacy, Shanghai Pudong Hospital, Fudan University Pudong Medical Center, Shanghai, China
[2] School of Pharmaceutical Sciences, Southern Medical University, Guangzhou, China
[3] Institute for Brain Research and Rehabilitation, South China Normal University, Guangzhou, China
* These authors contributed equally to this work.

Corresponding authors
Jingru Gong,
jingru_gong001@163.com
Shuaishuai Zhang,
zhangshuai2@smu.edu.cn

## ABSTRACT

Glycogen synthase kinase 3 (GSK-3) is a serine/threonine kinase widely expressed in various tissues and organs. Unlike other kinases, GSK-3 is active under resting conditions and is inactivated upon stimulation. In mammals, GSK-3 includes GSK-3 α and GSK-3β isoforms encoded by two homologous genes, namely, GSK3A and GSK3B. GSK-3β is essential for the control of glucose metabolism, signal transduction, and tissue homeostasis. As more than 100 known proteins have been identified as GSK-3β substrates, it is sometimes referred to as a moonlighting kinase. Previous studies have elucidated the regulation modes of GSK-3β. GSK-3β is involved in almost all aspects of brain functions, such as neuronal morphology, synapse formation, neuroinflammation, and neurological disorders. Recently, several comparatively specific small molecules have facilitated the chemical manipulation of this enzyme within cellular systems, leading to the discovery of novel inhibitors for GSK-3β. Despite these advancements, the therapeutic significance of GSK-3β as a drug target is still complicated by uncertainties surrounding the potential of inhibitors to stimulate tumorigenesis. This review provides a comprehensive overview of the intricate mechanisms of this enzyme and evaluates the existing evidence regarding the therapeutic potential of GSK-3β in brain diseases, including Alzheimer's disease, Parkinson's disease, mood disorders, and glioblastoma.

## INTRODUCTION

GSK-3 is an extensively conserved serine/threonine (S/T) protein kinase that catalyses the phosphorylation of threonine or serine residues, regulating numerous cellular biological processes, including glycogen metabolism, insulin signalling, cell growth, and differentiation (*Duda et al., 2020*). Inhibition of glycogen synthase (GS) by GSK-3 results in reduced glycogen synthesis in the liver and muscles, accompanied by elevated blood glucose levels or hyperglycaemia (*Gupte et al., 2020*). For this reason, GSK-3 is closely correlated with the pathogenesis and progression of many diseases, such as diabetes,

obesity, and cancer (*Amar, Belmaker & Agam, 2011*). GSK-3 is constitutively activated in resting cells and is frequently inhibited by growth factors (*e.g.*, insulin, insulin-like growth factor (IGF), vascular endothelial growth factor (VEGF), platelet-derived growth factor (PDGF), and nerve growth factor (NGF)) and other extracellular stimuli (*e.g.*, Wnt) (*Duda et al., 2018*). In humans, there are two isoforms of GSK-3, namely, GSK-3α (51 kDa) and GSK-3β (47 kDa), which are encoded by distinct genes, GSK3A and GSK3B, respectively. The catalytic domains of GSK-3α and GSK-3β proteins exhibit a remarkable 98% similarity, while their unique C-terminal regions diverge, showing only 36% homology (*Emma et al., 2020*). Although they have higher homology in the kinase domain, GSK-3α and GSK-3β have differentiated substrate preferences, and their cellular functions are at least partially nonredundant, as demonstrated by isoform-specific gene knockout (KO) studies in mice (*Gupte et al., 2022*). GSK-3α KO mice exhibit viability and normal developmental patterns, whereas GSK-3β KO leads to embryonic lethality, primarily attributed to severe liver degeneration (*Hoeflich et al., 2000*). The activity of the GSK-3β kinase depends on phosphorylation. Phosphorylation of tyrosine 216 (T216) in GSK-3β and tyrosine 279 (T279) in GSK-3α is needed for maximal activity, whereas phosphorylation of serine 9 (S9) in GSK-3β and serine 21 (S21) in GSK-3α results in inhibition of the kinase (*Moore et al., 2021*). For example, insulin inactivates GSK-3 by phosphorylation of specific residues, namely, serine 21 (p-Ser21-GSK-3α) and serine 9 (p-Ser9-GSK-3β), in a phosphatidylinositol 3-kinase (PI3K)-dependent manner (*Zakharova et al., 2019*). Among the many kinases that phosphorylate GSK-3β and thus inhibit its activity, the best known is protein kinase B (PKB or AKT), which acts downstream of PI3K and PDK1 kinase signalling. The inhibition of GSK-3β through phosphorylation is not exclusive to AKT but also involves other protein kinases, including protein kinase A (PKA) and protein kinase C (PKC) (*Ku et al., 2011*; *Moore et al., 2013*). In addition, GSK-3β regulation is influenced by various other regulators, including the mammalian target of rapamycin (mTOR), Wnt, and p38 mitogen-activated protein kinase (p38 MAPK) signalling pathways. These pathways also hold significant importance in the regulation of GSK-3β (*Golick et al., 2018*; *Pan & Valapala, 2022*; *Taelman et al., 2010*; *Thornton et al., 2008*). Despite this inhibitory phosphorylation, dephosphorylation of the Ser9 phosphate group on GSK-3β by protein phosphatases, such as protein phosphatase 2A (PP2A) and protein phosphatase 1 (PP1), enhances GSK-3β activity. This phenomenon has been observed in triple-negative breast cancer and neuronal degeneration (*Bennecib et al., 2000*; *Jian et al., 2022*; *Liang & Chuang, 2007*). Conversely, these regulators and protein phosphatases may also function as substrates for GSK-3β (*Hermida, Dinesh Kumar & Leslie, 2017*).

Over the past decades, dysregulation of GSK-3β has been linked to the pathogenesis of many disorders, including type 2 diabetes mellitus (T2DM), atherosclerosis, neurodegenerative diseases, and a variety of malignant tumours (*Lin et al., 2020*; *Yang et al., 2021*; *Zhang et al., 2018*). Because aberrant GSK-3β expression has been implicated in many diseases, accumulating studies have provided proof-of-concept that targeting GSK-3β is a promising strategy for treating various diseases. In tumours, GSK-3β has been demonstrated to play a paradoxical role as GSK-3β acts as a tumour promoter or

suppressor based on the cell type and phosphorylation status (*He et al., 2022*; *Li et al., 2017*; *Pecoraro et al., 2021*; *Zhang et al., 2020*). Furthermore, increasing evidence suggests that GSK-3β signalling plays a critical role in driving the advancement of neurodegeneration. Therefore, inhibition of GSK-3β is regarded as a prospective therapeutic strategy for central nervous system (CNS)-related disorders, especially Alzheimer's disease (AD) (*Eldar-Finkelman & Martinez, 2011*; *Lauretti, Dincer & Praticò, 2020*).

In the next section, we summarize the regulatory mechanisms of GSK-3β action, the physiopathological functions of GSK-3β in the brain, the current development of GSK-3β inhibitors, and treatment of brain and neurological diseases. The major focus of this review is on the intricate interactions between GSK-3β and neuroregulation, as well as the potential of GSK-3β for the therapeutic intervention of brain disease.

## Why this review is needed and who it is intended for

The brain is the main controller of learning, memory, movement, and other behaviours. Despite intensive research, the exact mechanisms that trigger brain disorders, such as Alzheimer's disease (AD) and Parkinson's disease (PD), are still not known, and at present, there is no cure for many brain diseases. Recently, several signalling molecules, especially glycogen synthase kinase-3β (GSK-3β), have been explored as possible candidate targets for the treatment of AD, PD, and mood disorders. This review discusses and summarizes the new advances in GSK-3β and its roles in brain physiology, neural development, neurodegenerative disorders, and neuropsychiatric disorders. Our review will appeal to researchers interested in the field of neuroscience, and it provides an overview of research on molecular targeted therapy for neurological and brain diseases. Although research on GSK-3β in the brain is still faced with many shortcomings and obstacles, the reported results in drug treatment, especially in GSK-3β inhibitors, are surprising. Importantly, these basic concepts and comprehensive knowledge are of great reference for chemists, biological scientists, pharmacists, and clinical workers as they may provide new insights into brain science and even a breakthrough in other fields.

## SEARCH METHODOLOGY

PubMed and Web of Science were used to identify relevant articles for this review, with the most recent search conducted in 2023. The search was performed in full-text journals, focusing on the most relevant advances in GSK-3β functions and their role in brain physiology and pathology. We categorized keywords, synonyms, and variants into categories, and used any combination of words from those categories for our search. The following categories were used for the search: (1) GSK-3β, which included GSK3β, glycogen synthase kinase-3β, and glycogen synthase kinase 3; and (2) brain disease, which included central nervous system, neural, cerebral, nervous, mind, Alzheimer's disease, Parkinson's disease, Huntington's disease, amyotrophic lateral sclerosis, multiple sclerosis, bipolar disorder, schizophrenia, glioblastoma, and ischaemia−reperfusion (I/R) injury. The words were merged *via* the Boolean operators "AND" and "OR". The initial search screened approximately 200 relevant articles written in English that could be useful for this review. No language restrictions were imposed.

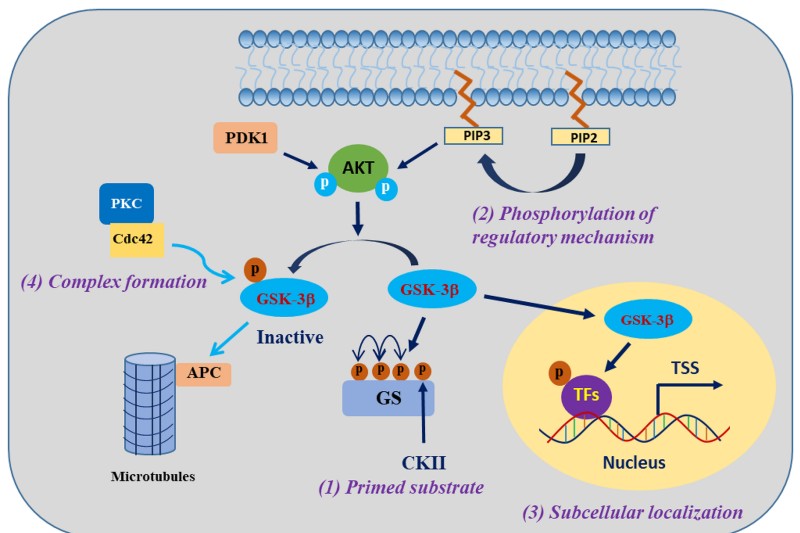

**Figure 1 Regulatory mechanisms governing the activities of GSK-3β.** GSK-3β phosphorylation of substrates is synergistically regulated by four mechanisms. Modified from *Jope, Yuskaitis & Beurel (2007)*. Copyright 2007, Springer Nature.

## Regulatory mechanisms of GSK-3β action

GSK-3β is involved in the regulation of cell biological functions through specific substrate phosphorylation. The regulation of GSK-3β activity encompasses the following four pivotal mechanisms: autophosphorylation regulation of GSK-3β, subcellular localization of GSK-3β, assembly of protein complexes harbouring GSK-3β, and phosphorylation status of GSK-3β substrates (*Jope & Johnson, 2004*) (Fig. 1). One of the clearest regulatory mechanisms is the inhibition of GSK-3β activity by phosphorylation of Ser9 in GSK-3β (corresponding to Ser21 in GSK-3α) (*Patel & Werstuck, 2021*). The PI3K/AKT signalling pathway is the primary regulator of GSK-3β, which is activated in response to insulin and several growth factors. However, AKT (PKB) and other kinases, including protein kinase A (PKA), ribosomal 70-kDa protein S6 kinase (p70S6K), and p90 ribosomal S6 kinase (p90RSK), phosphorylate the inhibitory Ser9 residue of GSK-3β, which results in the inactivation of GSK-3β (*Cervello et al., 2017*). In contrast, phosphorylation of Tyr216 in GSK-3β (corresponding to Tyr279 in GSK-3α) by Src, FYN, and PYK2 enhances the catalytic activity of GSK-3β kinase, which may be constitutive in resting cells, but the mechanism regulating this modification is unclear (*Bhat et al., 2000*; *Nagini, Sophia & Mishra, 2019*).

The activity of GSK-3β is subject to regulation through subcellular localization, particularly within the nucleus and mitochondria, where GSK-3β exhibits heightened activity and undergoes dynamic regulation (*Bijur & Jope, 2003*). When translocated into the nucleus, GSK-3β exhibits phosphorylation activity towards a diverse range of substrates, mostly known as transcription factors (TFs) or epigenetic regulatory factors, such as RXRα, p53, and KDM1A (*Eom & Jope, 2009*; *Zhang et al., 2020*; *Zhou et al., 2016*). The localization of GSK-3β within protein complexes either facilitates or impedes its activity towards specific substrates. For example, in the canonical Wnt signalling pathway,

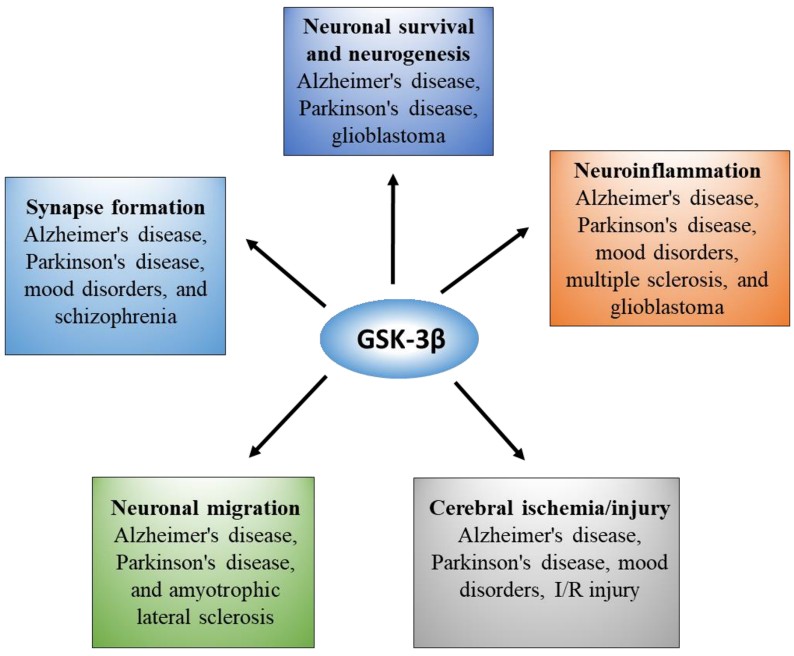

**Figure 2 Overview of GSK-3β functions and correlated diseases in the brain.**

GSK-3β and the β-catenin transcriptional coactivator coexist on the Axin scaffold protein, and this colocalization guides GSK-3β to phosphorylate β-catenin, consequently triggering the degradation of β-catenin protein in the cytosol. Activation of the Wnt signalling pathway prevents GSK-3β from accessing β-catenin, resulting in the accumulation of active β-catenin (*Marineau, Khan & Servant, 2020*). The fourth mechanism refers to the effect of GSK-3β being regulated by the substrate phosphorylation state, which is a mechanism that indirectly regulates the substrate phosphorylation efficiency of GSK-3β. Most of the substrates of GSK-3β must be in the "primed" (sensitized) state. Under the condition that GSK-3β is prephosphorylated at a residue four amino acids C-terminal to the phosphorylation site by another "priming kinase" (activated kinase), GSK-3β further phosphorylates the "primed" substrate; that is, the regulation of substrate phosphorylation activity by GSK-3β usually requires the coordination of the "priming kinase" (*Duan et al., 2022*; *Jope, Yuskaitis & Beurel, 2007*). These complex mechanisms of GSK-3β regulation provide GSK-3β with specific control over its substrates, a particularly important ability of the enzyme that enables GSK-3β to phosphorylate numerous substrates and thus regulate many biological functions.

## Multifaceted functions of GSK-3β in the brain

GSK-3β is the most abundant in the brain and nervous system, and its expression level increases with age (*Lee et al., 2006*). GSK-3β not only mediates abnormal pathological processes but is also essential for the normal physiological function of the brain. To obtain a comprehensive understanding, we summarized the pleiotropic role of GSK-3β in the brain in Fig. 2.

### GSK-3β, synaptic plasticity, and memory formation

The hippocampus is widely recognized as the central hub for learning and memory within the brain, and it plays a crucial role in synaptic plasticity, encompassing the mechanisms underlying both long-term potentiation (LTP) and long-term depression (LTD). LTP refers to the enduring enhancement of synaptic strength, while LTD represents the contrasting process (*Parvez et al., 2023*). GSK-3β exhibits high expression levels in the hippocampus of a healthy brain where it assumes a crucial function in synaptic plasticity and the formation of memories. *Liu et al. (2017)* reported that mice with GSK-3β deletion specifically in the excitatory neurons of the dentate gyrus (DG) exhibit impairments in spatial and fear memory. Subsequent investigations have revealed that deletion of GSK-3β in the DG subset leads to the inhibition of synaptic transmission within the hippocampus; additionally, it results in decreased expression levels of GluN1, GluN2A, GluN2B, GluA1, PSD93, drebrin, and synaptophysin (*Liu et al., 2017*). Consistently, *Koike et al. (2021)* demonstrated that aged GSK-3β$^{+/-}$ mice manifest deficiencies in the formation of both short-term and long-term memories, thereby offering additional evidence for the essential role of GSK-3β in memory formation during advanced age. Mechanistically, *Peineau et al. (2007)* reported that the activation of GSK-3β is enhanced during NMDA receptor-dependent LTD *via* activation of PP1, while the activity of GSK-3β is inhibited by the induction of LTP. The induction of LTP in both the DG and CA1 regions of the hippocampus leads to an elevation in the inhibitory phosphorylation of GSK-3β at Ser9 (*Cunningham et al., 2017*).

Memory impairment in old age is a hallmark of AD, with dementia developing in the final stages. Alzheimer's disease is marked by β-amyloid (Aβ) deposits and neurofibrillary tangles (NFTs). Detectable NFTs in AD-associated memory impairment and dementia are associated with synaptic and neuronal loss in diseased brains (*Ashrafian, Zadeh & Khan, 2021*). Prior to NFT formation, tau undergoes hyperphosphorylation due to the activation of GSK-3β, resulting in the formation of tau oligomers with a granular structure. The hyperphosphorylated tau is strongly linked to the loss of synapses and the impairment of memory induced by Aβ. Mice overexpressing GSK-3β show an accumulation of hyperphosphorylated tau, a reduction in hippocampal LTP, and memory impairment in object recognition tests (*Takashima, 2012*). Notably, this memory deficit in mice is reversed when tau expression stops. Consequently, reducing tau levels and inhibiting GSK-3β each rescue memory impairment in AD models (*Gómez de Barreda et al., 2010*; *Roberson et al., 2007*). Another possible mechanism is that GSK-3β inactivation enables stabilization of β-catenin, which protects β-catenin from degradation mediated by the proteasome. β-catenin is expressed in both pre- and postsynaptic terminals, and it is responsible for cell adhesion and synaptic structure (*Hui et al., 2018*; *Murase, Mosser & Schuman, 2002*). Downregulation of the β-catenin signalling pathway due to increased GSK-3β activity has been observed in AD brains (*Vallée et al., 2018*). Given the pivotal role of GSK-3β in the regulation of synaptic plasticity and memory formation, the reduction in GSK-3β activity obtained by pharmacological approaches has been extensively explored and reported in several AD models (*Iqbal et al., 2023*).

### GSK-3β and neuroinflammation

Inflammation has become a dominant contributor to brain disease, with principal participation from ageing and proteinopathy, which has been widely studied to understand its mechanism in neural pathology (*Samim Khan et al., 2023*). The initial defensive reaction of the organism in response to allergic stimuli involves immune-mediated inflammation. Although the initial response is protective, the persistent and excessive response causes pathological lesions. In the neural network, the equilibrium between inflammatory and anti-inflammatory processes is upheld through the activation of M1 and M2 microglia. M1 microglia exhibit the capacity to generate inflammatory factors primarily mediated by Toll-like receptors (TLRs), whereas M2 microglia function as reparative agents, primarily releasing IL-10 and IL-4 (*Cherry, Olschowka & O'Banion, 2014*). As a result of pathological stimuli within the brain, the innate immune response is initially engaged, leading to the subsequent transformation of microglia-mediated innate immunity into neuroinflammation. Neuroinflammation is associated with glial activation, blood–brain barrier (BBB) breakdown, proinflammatory cytokine release, and leukocyte invasion, in which the molecular mechanism generally involves the activity of NOD-like receptor family pyrin domain-containing protein 3 (NLRP3), Toll-like receptors (TLRs), and nuclear transcription factors, such as nuclear factor kappa B (NF-kB), nuclear factor of activated T cells (NFAT), cAMP response element-binding protein (CREB), and signal transducer and activator of transcription 3 (STAT3) (*Candelario-Jalil, Dijkhuizen & Magnus, 2022*; *Jurcau & Simion, 2021*; *Zhu et al., 2021*).

NLRP3 is a cytoplasmic pattern recognition receptor (PRR) that is prominently expressed in macrophages, and it serves as a crucial constituent of the inflammasome. In the past decade, NLRP3 has been shown to be involved in neuroinflammatory processes, in which the interplay between the innate immune system and the caspase-1 apoptotic protein plays a significant role (*Voet et al., 2019*). The assembly of the NLRP3 inflammasome results in the secretion of potent proinflammatory cytokines, namely, IL-1β, IL-18, and IL-33. These inflammatory factors play a crucial role in neurodegeneration. After the discovery of the regulatory role of GSK-3β in the brain, its participation in NLRP3-mediated inflammatory pathways has been widely explored. Agents capable of inducing the phosphorylation of GSK-3β to mitigate NLRP3-mediated inflammation, oxidative stress, apoptosis, and autophagy have been assessed in various models of neurological disorders (*Li et al., 2021*; *Liu et al., 2020b*; *Wang et al., 2021*, *2019*). In the brain, Toll-like receptors (TLRs) mainly modulate glial and neuronal functions, as well as innate immunity and neuroinflammation, under physiological or pathological conditions (*Fei et al., 2022*; *Mowry et al., 2021*; *Schilling et al., 2021*). During the past decade, numerous studies have demonstrated that GSK-3β is a key regulator of the TLR signalling pathway *via* TLR3 and TLR4 (*Ko & Lee, 2016*). TNFR-associated factors (TRAFs) are critical for the production of inflammatory cytokines and antiviral responses in TLR3-mediated signalling pathways. *Ko et al. (2015)* identified TRAF6 as a direct E3 ligase for GSK-3β, and they demonstrated that TRAF6-mediated GSK-3β ubiquitination is essential for cytokine production by promoting complexes assembled by TRIF, a TLR3

adaptor protein. These researchers also reported that Src phosphorylation is regulated by GSK-3β *via* TNFR-associated factor 2 (TRAF2)-mediated Src ubiquitination. Deficits of GSK-3β in mouse embryonic fibroblasts significantly reduce IFN-stimulated gene expression due to a reduction in Src tyrosine phosphorylation (*Ko et al., 2019*). Compared to the wild type (WT) group, TLR3 deficiency significantly inhibits programmed necrosis of brain cells in neonatal mice (*Zhang et al., 2022a*). TLR3 serves as a vital modulator of neuronal survival and developmental neuroplasticity. TLR3-deficient mice display augmented volumes of the hippocampal CA1 and DG regions, along with heightened levels of the GluR1 AMPA receptor subunit in the CA1 region (*Okun et al., 2010*). In TLR4 knockout mice, stress-activated GSK-3β activity and the production of hippocampal cytokines and chemokines are attenuated. Similarly, administration of TDZD-8, a GSK-3β inhibitor, significantly reduces the expression of most hippocampal cytokines and chemokines (*Cheng et al., 2016*). In addition, TDZD-8 treatment downregulates TLR4 protein expression, upregulates claudin5 protein expression, and significantly improves cognitive function in aged mice (*Liang et al., 2020*).

Different transcription factors, such as nuclear factor kappa B (NF-kB), c-Jun, nuclear factor of activated T cells (NFAT), cAMP response element-binding protein (CREB), and signal transducer and activator of transcription 3 (STAT3), have been shown to be substrates for GSK-3β activity (*Gao et al., 2017*; *Götschel et al., 2008*; *Lakshmanan et al., 2015*). Furthermore, it is well documented that these GSK-3β substrates play important roles in the regulation of neuroinflammation (*Golpich et al., 2015*). Among these factors, NF-kB is a principal transcription factor activated by inflammatory processes leading to cytokine production. A previous study has indicated that NF-kB is regulated by GSK-3β at the level of the transcriptional complex (*Hoeflich et al., 2000*). In addition, GSK-3β directly phosphorylates the NF-kB protein at Ser468 in unstimulated cells, thereby controlling the basal activity of NF-kB (*Buss et al., 2004*). NF-kB is expressed in microglia, neurons, and astrocytes, and it plays an important role in the regulation of inflammatory intermediates during neuronal dysfunction (*Singh et al., 2020*). NFAT activity is needed for the A-stimulated microglial activation that occurs during Alzheimer's disease (*Rojanathammanee et al., 2015*). Activation of STAT3 and the c-Jun signalling pathway may contribute to neuroinflammation and cognitive impairment (*Hu et al., 2021b*; *Li et al., 2022*). However, activation of the CREB signalling pathway significantly increases the survival of neurons, improves cognitive behaviour, and promotes anti-inflammatory effects (*Sharma & Singh, 2020*).

### GSK-3β, neuronal survival, and neurogenesis

Neurogenesis involves cell survival, proliferation, and differentiation, which are strictly controlled by epigenetic mechanisms of gene transcription. Neurogenesis is traditionally recognized as an embryogenic phenomenon, but numerous studies have suggested that this process exists in certain areas of the brain (*Espinós et al., 2022*). In recent years, neurogenesis has attracted much attention, as drugs and targets promoting neurogenesis have shown beneficial results in the treatment of neurodegenerative diseases. In support of the hypothesis that GSK-3β is an important regulator in neurogenesis, transgenic mice

overexpressing GSK-3β exhibit a reduction in proliferation and maturation of new functional DG neurons, as well as a severe memory impairments. This GSK-3β-dependent depletion of neurogenesis contributes to microglial activation and Aβ-mediated neuronal death, further promoting neurodegeneration (*Hernandez, Lucas & Avila, 2013*). GSK-3β inhibition has been shown to enhance the proliferation, migration, and differentiation of neural stem cells *in vitro*. Inhibition of GSK-3β with the small molecule, tideglusib, induces neurogenesis in the DG of the hippocampus *in vivo* (*Morales-Garcia et al., 2012*). The above findings indicate that inactivation of GSK-3β is beneficial to promote the survival of functional neurons in neuropathology with impaired neurogenesis.

A deficit in hippocampal neurogenesis contributes to cognitive decline in old age. Activated GSK-3β accelerates hippocampal neurogenesis at the early stage, while pharmacological inhibition of GSK-3β is efficient in preserving hippocampal neurogenesis in senescent mice (*Liu et al., 2020a*). Inflammation is also implicated in the neuropathology of neurodegenerative disorders, as increased levels of proinflammatory cytokines, such as interleukin-1β (IL-1β), have been shown to be detrimental to hippocampal neurogenesis. Previous results have suggested that GSK-3β activation is involved in the antiproliferative and progliogenic effects of IL-1β, and reduced GSK-3β activity facilitates the restoration of hippocampal neurogenesis in neuroinflammatory conditions (*Green & Nolan, 2012*). Glucose is indispensable for neuronal survival, and even minor changes in the glucose supply may result in serious consequences to the normal physiology of the nervous system. Thus, glucose homeostasis needs to be precisely regulated for the normal survival of neurons. Given the pivotal roles of GSK-3β in insulin sensitivity, glycogen synthesis, and glucose metabolism, GSK-3β is also necessary for neurogenesis.

### GSK-3β and neural migration

Considerable evidence shows that GSK-3β directly or indirectly regulates neuronal migration in the nervous system. Several studies have investigated GSK-3β targeting the APC protein. Apart from its involvement in protein degradation, APC is a cytoskeletal protein that relies on microtubules and contributes to the maintenance of the polarized glial scaffold during brain development (*Barth, Caro-Gonzalez & Nelson, 2008*; *Fang & Svitkina, 2022*). In the absence of APC, glial cells lose their polarity and respond to extracellular polarity signals, such as neuregulin-1. Conditional gene targeting to eliminate APC further induces instability in the glial microtubule cytoskeleton, thereby indicating a significant role of APC in neuronal migration (*Yokota et al., 2009*). It remains unclear whether the migration defects are caused by the disruption of the glial scaffold or the absence of microtubule-associated APC in cortical neurons.

GSK-3β has been implicated in the regulation of neuronal migration through its influence on β-catenin. Mouse studies have provided evidence that genetic manipulation of β-catenin, either through deletion or overexpression, disrupts the development of the brain and spinal system (*Zechner et al., 2003*). The migration process primarily arises from aberrant neural progenitor development. In the context of a mutated nervous system, the presence of abnormally localized neurons has also been observed. Additionally, the levels
of β-catenin in progenitor cells influence the positioning of neurons (*Mutch et al., 2009*). A possible explanation for the abnormal positioning of neurons is delayed determination of progenitor fate. However, studies on β-catenin mutations provide strong evidence for the potential involvement of GSK-3β in neuronal migration. *Morgan-Smith et al. (2014)* demonstrated that GSK-3β is essential for radial migration and dendritic orientation. Interestingly, this GSK-3β regulation of migration in neurons is independent of β-catenin signalling (*Morgan-Smith et al., 2014*).

Researchers have also investigated the molecular regulation of GSK-3β in the migration of neurons. For example, DISC1, a prominent susceptibility factor for numerous mental disorders, governs the process of neuronal migration during brain development and has been implicated in the regulation of neuronal migration through its interaction with GSK-3β. At midembryonic stages, when neural progenitor proliferation is active, DISC1 interacts with GSK-3β. However, at later embryonic stages, when neuronal proliferation is occurring, DISC1 dissociates from GSK-3β (*Ishizuka et al., 2011*). LKB1, an evolutionarily conserved polarity kinase also known as the upstream kinase of AMPK, is another important regulator of neuronal migration in the development of neurons. *Asada & Sanada (2010)* reported that LKB1 phosphorylates GSK-3β at Ser9, thereby promoting neuronal migration in the developing neocortex. The delineation of the roles and underlying mechanisms of GSK-3β in neuronal migration holds paramount importance, as abnormalities in neuron migration are implicated in various neural disorders.

### GSK-3β and cerebral ischaemia

The brain, being the most vulnerable organ to hypoxia and ischaemia, is highly susceptible to the detrimental effects of cerebral ischaemia. This condition poses a significant threat to human life because it can lead to severe impairment of brain functions. Additionally, both cerebral ischaemia and the subsequent reperfusion phase inflict extensive damage to the brain. Numerous studies have indicated the pivotal involvement of GSK-3β in neuronal injury following cerebral ischaemia. Most recently, *Peng et al. (2022)* observed downregulation of AKT accompanied by activation of GSK-3β within 12 h in cerebral ischaemia/reperfusion, and they suggested this downregulation is mediated by mTORC instead of PI3K and PDK1 signalling. Downregulation of GSK-3β using a siRNA-mediated approach markedly attenuates neuronal damage and enhances cell viability in rat models of ischaemia/reperfusion. Conversely, upregulation of GSK-3β significantly exacerbates neurological impairments and inflicts damage upon cerebral cortical neurons in ischaemia/reperfusion models (*Li et al., 2016*). In line with the findings of *Kisoh et al. (2017)* experimental models have demonstrated a substantial increase in the phosphorylation of GSK-3β at Ser9, as well as the phosphorylation of AKT, on the seventh day following cerebral ischaemia. Additionally, administration of a PI3K inhibitor results in a reduction in AKT activation and phosphorylation of GSK-3β at Ser9 in response to cerebral ischaemia (*Kisoh et al., 2017*). *Chen et al. (2017)* examined the expression of GSK-3β and p-GSK-3β in the gerbil hippocampal CA1 area after transient cerebral ischaemia, and they reported that p-GSK-3β is highly expressed in astrocytes located in the stratum oriens and radiatum. The results indicate an increase in GSK-3β immunoreactivity in

pyramidal cells of the CA1 region at 6 h following ischaemia–reperfusion. However, GSK-3β levels decrease after 12 h and are scarcely detectable in CA1 pyramidal cells at 5 days postischaemia–reperfusion. Furthermore, p-GSK-3β is slightly decreased in CA1 pyramidal cells at 6 and 12 h but significantly increased at 1 and 2 days, and it is barely detectable in CA1 pyramidal cells at 5 days after ischaemia–reperfusion (*Chen et al., 2017*).

The occurrence of ischaemia–reperfusion (I/R) injury can lead to neuronal cell death, and it is closely related to oxidative stress. Nuclear factor erythroid 2-related factor 2 (Nrf2) is a master regulator of oxidative stress correlated with brain I/R injury. Several studies have suggested that activation of Nrf2 *via* GSK-3β modulation alleviates I/R-induced apoptosis and oxidative stress in neurons (*Bai et al., 2021*; *Duan et al., 2019*; *Liao et al., 2020*; *Xu et al., 2021*). The involvement of GSK-3β in ischaemia–reperfusion models can be attributed to its regulation of the Nrf2/ARE pathway, resulting in a reduction in oxidative stress. Treatment with a GSK-3β inhibitor or GSK-3β siRNA has been shown to prevent neuronal injury caused by brain ischaemia through the activation of the Nrf2 signalling pathway (*Li et al., 2016*; *Pang et al., 2016*). In support of the hypothesis that GSK-3β is an important regulator in cerebral ischaemia, inhibition of GSK-3β using GSK-3β inhibitors enables protection of the brain from injury in various ischaemia–reperfusion models (*Gao, Yang & Cui, 2021*; *Wang et al., 2017*).

Altogether, GSK-3β is widely involved in physiological and pathological processes of the brain with sophisticated regulatory mechanisms. It is imperative for future investigations to elucidate additional facets of GSK-3β signalling in cerebral development that could hold therapeutic significance in the realm of brain disorders.

### GSK-3β and its inhibitors in brain diseases

Due to its multifaceted functions in the brain, GSK-3β has been strongly implicated in a variety of brain diseases, including Alzheimer's disease, Parkinson's disease, mood disorders, schizophrenia, and glioblastoma (GBM). Substantial evidence suggests that small molecule inhibitors of GSK-3β have great value in drug development. Generally, GSK-3β inhibitors are classified into the following four categories: (1) cations encompassing the mood stabilizer lithium, along with other metal ions, such as zinc and copper, which exert inhibitory effects on GSK-3β at millimolar or submicromolar concentrations, respectively; (2) ATP competitive inhibitors; (3) allosteric non-ATP competitive inhibitors; and (4) substrate competitive inhibitors (SCIs). The representative GSK-3β inhibitors and their reported effects in brain diseases are summarized in Table 1.

To date, researchers have invested a great deal of effort into developing GSK-3β inhibitors as potential drugs, especially for the treatment of neurodegenerative and psychiatric disorders (*Arciniegas Ruiz & Eldar-Finkelman, 2021*). For example, AF3581 is a new class of GSK-3β inhibitor, and studies have shown that this compound is effective in chronic mild stress-induced depression (mimicking the low phase of bipolar disorder) and mouse aggression (mimicking the high phase), indicating the therapeutic potential of GSK-3β inhibitors in treating patients with bipolar disorder (*Capurro et al., 2020*). Previous studies have documented the efficacy of Schisandrin B stereoisomers as inhibitors of GSK-3β. These stereoisomers have demonstrated the ability to substantially enhance the

**Table 1 Representative GSK-3β inhibitors and its usage in brain disease models.**

| Type | Typical compounds | Structures | Applications in brain disease |
|---|---|---|---|
| ATP competitive | SB-216763, SB-415286 | | AD, PD, BD, I/R injury (*Kalinichev & Dawson, 2011*; *Wang et al., 2019*; *Xiong et al., 2013*; *Zhang, Yin & Zhang, 2014*) |
| | CHIR99021, CHIR98023 | | HD, GBM, I/R injury (*Hu et al., 2021a*; *Yang et al., 2020*; *Zhang et al., 2022b*) |
| | AZD1080 | | PD (*Hu et al., 2020*) |
| | TWS119 | | AD, I/R injury (*Gao, Yang & Cui, 2021*; *Jiang et al., 2021*) |
| Non-ATP competitive | TDZD-8 | | AD, PD, BD, GBM, MS, SZ, I/R injury (*Aguilar-Morante et al., 2010*; *Beurel et al., 2013*; *Duka et al., 2009*; *Huang et al., 2017*; *Kalinichev & Dawson, 2011*; *Koehler, Shah & Williams, 2019*; *Willi et al., 2013*) |
| | Tideglusib | | AD, PD, ALS, BD, GBM, I/R injury (*Al-Zaidi et al., 2021*; *Bahmad et al., 2021*; *del Ser et al., 2013*; *Martínez-González, Gonzalo-Consuegra & Gómez-Almería, 2021*; *Moretti, 2015*; *Wang et al., 2016*) |

(Continued)

| Type | Typical compounds | Structures | Applications in brain disease |
|---|---|---|---|
| Substrate competitive | L803mts |  | AD, MS, HD (*Beurel et al., 2013*; *Eldar-Finkelman & VanHook, 2016*; *Rippin et al., 2021*) |
| | ITDZs |  | MS (*Palomo et al., 2012*; *Redondo et al., 2012*) |
| Cations | Lithium | Li | AD, PD, ALS, MS, BD, GBM, SZ, I/R injury (*Leucht, Kissling & McGrath, 2007*; *Ochoa, 2022*; *Smith, Mill & Lunnon, 2020*; *Young, 2009*) |

**Note:**

Abbreviations of disease: AD, Alzheimer's disease; PD, Parkinson's disease; HD, Huntington9s disease; ALS, Amyotrophic Lateral Sclerosis; MS, Multiple Sclerosis; BD, Bipolar Disorder; SZ, Schizophrenia; GBM, Glioblastoma; I/R injury, Ischemia-reperfusion (I/R) injury.

expression of p-GSK-3β (Ser9) while reducing the expression of p-GSK-3β (Tyr216 and Tyr279). Furthermore, they have shown promising effects in mitigating cell damage induced by amyloid β (Aβ) and ameliorating cognitive impairment in mice with Alzheimer's disease (AD). These findings highlight the potential utility of these stereoisomers as neuroprotective agents for the treatment of Alzheimer's disease (*Hu et al., 2019*). After intranasal administration of 1-methyl-4-phenyl-1,2,3,6-tetrahydropyridine (MPTP), rodents display time-dependent impairments in emotion, cognition, and motor function, thus mimicking Parkinson's disease in rodents. Lithium salt, a GSK-3β inhibitor, prevents olfactory discrimination and short-term memory impairment in an intranasal MPTP-induced Parkinson's disease rat model (*Castro et al., 2012*).

### GSK-3β and Alzheimer's disease

Alzheimer's disease (AD) is a neurodegenerative disease, a type of late-onset dementia, characterized by gradual loss of memory, loss of speech ability, cognitive impairment, and other mental symptoms. AD is characterized by profound alterations in the morphology of numerous neuronal somata and axon terminals. The typical neuropathological changes of AD include the formation of senile plaques, which are deposits of Aβ protein in the brain, neurofibrillary tangles formed by the excessive phosphorylation of tau protein, neuronal apoptosis, and a series of inflammatory reactions (*Chou et al., 2012*). Neurofibrillary degeneration (NFD) is a type of intracellular damage caused by the modification of tau protein, which is a structural protein involved in stabilizing the neuronal cytoskeleton. During AD, tau protein undergoes hyperphosphorylation, which causes it to detach from

microtubules and form paired pathological helical filaments, leading to progressive neurofibrillary degeneration.

GSK-3β is significantly expressed in the neuronal cell bodies of the brains of AD patients (*Pei et al., 1997*). Studies have shown that GSK-3β is a key enzyme for the hyperphosphorylation of tau protein. Overexpression of GSK-3β in the brains of transgenic mice leads to the hyperphosphorylation of tau protein, microtubule breakdown, and the appearance of neurofibrillary degeneration (*Brownlees et al., 1997*; *Hernandez, Lucas & Avila, 2013*). The activity of GSK-3β contributes to the production of Aβ and Aβ-mediated neuronal death. Mechanistically, tau protein is a substrate for GSK-3β phosphorylation and p-tau enable to form paired pathological helical filaments, leading to progressive neurofibrillary degeneration. GSK-3β is involved in regulating the protein hydrolysis and cleavage of amyloid precursor protein (APP), as well as the production of Aβ, which is a small toxic peptide. Aβ is derived from the protein hydrolysis of APP, and inhibiting GSK-3β reduces the production of Aβ and protects neurons from the toxic effects of the peptides. In turn, abnormal aggregation of Aβ increases the activity of GSK-3β, establishing a mechanistic link between the two major hallmarks of AD (accumulation of Aβ and NFD) (*Qing et al., 2008*; *Terwel et al., 2008*). Studies have found that inhibiting the activity of GSK-3β reduces the production of Aβ and tau phosphorylation while improving learning and memory abilities (*Zhang et al., 2011*). Therefore, GSK-3β is considered a potential target for treating Alzheimer's disease. Currently, some GSK-3β inhibitors have entered preclinical studies, but their efficacy and safety still need further verification (*Arciniegas Ruiz & Eldar-Finkelman, 2021*).

### GSK-3β in Parkinson's syndrome

Parkinson's disease (PD) is a chronic neurodegenerative movement disorder, and it is the second most common neurodegenerative disease. PD is characterized by the degeneration of substantia nigra dopaminergic neurons and depletion of dopamine in the striatum, as well as the abnormal accumulation of α-synuclein and synphilin-1 (α-synuclein-interacting protein), resulting in pathological and clinical abnormalities (*Jankovic & Tan, 2020*). Dopamine helps to guide muscle activity, and when there is a significant loss of dopaminergic neurons in the substantia nigra of the brain, characteristic movement difficulties of PD, such as tremors, rigidity, bradykinesia, impaired balance, and impaired coordination, can occur (*Reich & Savitt, 2019*).

Numerous studies have found that the expression of GSK-3β is higher in the brains of patients with PD. In the striatum of PD patients after death, the levels of α-synuclein and p-GSK-3β phosphorylated at Tyr 216 are higher, and activated GSK-3β hyperphosphorylates tau to produce toxic pathological forms of p-tau (*Wills et al., 2010*). In transgenic mouse models, the activation of GSK-3β depends on the presence of α-synuclein, and overexpression of α-synuclein is associated with increased activity of GSK-3β, indicating that the accumulation of α-synuclein contributes to the activation and increase of GSK-3β in the brain of PD, leading to excessive phosphorylation of tau. Both *in vitro* and *in vivo*, GSK-3β is activated by MPTP in a strictly α-synuclein-dependent manner and is responsible for the excessive phosphorylation of tau protein (*Duka et al., 2009*).

Mechanistically, α-synuclein is a substrate for GSK-3β phosphorylation and has been reported to activate NLRP3 inflammasome assembly by TLR2 stimulation. Interestingly, α-synuclein reversely leads to GSK-3β activation, and an acidic region of α-synuclein is responsible for the stimulation of GSK-3β-mediated tau phosphorylation (*Samim Khan et al., 2023*). Thus, it has been concluded that there is crosstalk between α-synuclein and GSK-3β-mediated tau phosphorylation, resulting in PD progression. A previous study has reported that heat shock protein 70 (Hsp70) suppresses the α-synuclein-induced phosphorylation of tau by GSK-3β through direct binding to α-synuclein, suggesting that Hsp70 may act as a novel therapeutic target to counteract α-synuclein-mediated tau phosphorylation in PD (*Kawakami et al., 2011*).

Inhibition of GSK-3β reduces the accumulation of α-synuclein and alleviates the excessive phosphorylation of tau protein (*Duka et al., 2009*). Synphilin-1 is an α-synuclein-interacting protein, and GSK-3β specifically phosphorylates synphilin-1, reducing its ubiquitination *in vitro* and *in vivo*, which reduces its degradation by proteasomes (*Avraham et al., 2005*). A small amount of GSK-3β is also detected in mitochondria, and studies have shown that increased mitochondrial GSK-3β activity enhances the production of reactive oxygen species and disrupts mitochondrial morphology. Chemical inhibitors of GSK-3β inhibit cell apoptosis induced by rotenone, an inhibitor of complex I of the electron transport chain, and attenuate GSK-3β-mediated damage to mitochondria (*King et al., 2008*). Inhibition of GSK-3β activity is a key element in the treatment of PD, as GSK-3β appears to be a specific therapeutic target for PD due to its involvement in neuronal apoptosis, phosphorylation of tau protein, aggregation of α-synuclein, aggregation of synphilin-1, and mitochondrial dysfunction (*Golpich et al., 2015*).

### GSK-3β in bipolar disorder

Bipolar disorder (BD), also known as manic-depressive illness, is a severe mental disorder. The symptoms of bipolar disorder first appear in adolescence or early adulthood, and they recur unpredictably with features of cyclic alternation between manic and depressive moods, characterized by periodic disturbances in mood, energy patterns, and behaviour (*Vieta et al., 2018*). Lithium and valproic acid are representative drugs and mood stabilizers used for bipolar disorder, which can alleviate mild to moderate manic episodes in BD patients (*Scarselli, 2023*). Both lithium and valproic acid exert therapeutic effects by inhibiting GSK-3β (*Dandekar et al., 2018*). Drugs used to treat bipolar disorder usually inhibit GSK-3β, indicating that GSK-3β plays a critical role in the therapeutic effects of bipolar disorder treatment.

GSK-3β is not only a target of lithium but also of other categories of mood stabilizers, antidepressants, and antipsychotic drugs (*Beaulieu, 2007*). In transgenic mice overexpressing GSK-3β, research has shown that the overexpression of GSK-3β leads to hyperactivity and ADHD-like symptoms, similar to human mania (*Prickaerts et al., 2006*). Heterozygous GSK-3β mice exhibit normal morphology, reduced exploratory activity, and a lithium-mimetic antidepressant-like state (*O'Brien et al., 2004*). Increasing evidence supports the important role of GSK-3β in the treatment of mania and depression (*Dandekar et al., 2018*). GSK-3β represents a promising therapeutic target for addressing

depression, with demonstrated antidepressant effects observed through the administration of GSK-3β inhibitors in animal models (*Rosa et al., 2008*). Increased GSK-3β kinase activity has been detected in brain samples of human patients with bipolar disorder (BD), and abnormal GSK-3β activity occurs in patients with severe depression, indicating that abnormal GSK-3β activity promotes the onset of BD (*Polter et al., 2010*). Dysregulation of serotonin (5-HT) neurotransmission in the brain is considered the basis of bipolar disorder. In transgenic mice, the decrease in brain 5-HT levels is accompanied by the activation of GSK-3β, and the inactivation of GSK-3β alleviates abnormal behaviour caused by 5-HT deficiency. Various 5-HT drugs inhibit brain GSK-3β signalling, indicating that targeting the GSK-3β signalling pathway may provide a strategy for the treatment of certain 5-HT-related psychiatric disorders (*Beaulieu et al., 2008*; *Latapy et al., 2012*; *Zheng et al., 2021*).

A considerable array of GSK-3β inhibitors has been examined in models of bipolar disorder (BD). In mouse models of mania, administration of various GSK-3β inhibitors, including TDZD-8, SB-216763, SB-627772, AF3581, and indirubins, mitigates the "manic" effects, as evidenced by decreased hyperactivity and restoration of normal ambulation behaviour. GSK-3β inhibitors also demonstrate notable therapeutic advantages in addressing the depressive phase in animal models. Administration of tideglusib, VP2.51, and the SCI peptide yields antidepressant-like outcomes (*Arciniegas Ruiz & Eldar-Finkelman, 2021*).

### GSK-3β and schizophrenia

Schizophrenia is a common debilitating neurological disorder with a multigenetic basis, and it is characterized by poor emotional response, impaired language reasoning, and significant social dysfunction (*Zamanpoor, 2020*). AKT1 and GSK-3β play critical roles in synaptic plasticity and neuronal function in the central nervous system. The AKT/GSK-3β signalling pathway is involved in schizophrenia, with AKT1 being a potential susceptibility gene for schizophrenia. Cognitive impairment, abnormal synaptic morphology, neuronal atrophy, and dysfunction of neurotransmitter signalling may partly be explained by reduced PI3K/AKT signalling in schizophrenia. Furthermore, lower levels of AKT may have a detrimental effect on neurodevelopment by increasing the effect of risk factors, attenuating the effect of growth factors, and reducing the response of patients to treatment of antipsychotic agents (*Zheng et al., 2012*). AKT1-3 are upstream inhibitors of GSK-3β, and Ser9 in GSK-3β is a phosphorylation site for AKT. Phosphorylation of GSK-3β (Ser9) leads to its inactivation. It has been reported that the protein level of AKT1 is reduced in patients with schizophrenia, accompanied by a decrease in phosphorylation of Ser9 in GSK-3β and an increase in GSK-3β activity (*Emamian et al., 2004*). GSK-3β has hundreds of single nucleotide polymorphisms (SNPs), and it has been reported that individuals carrying the C allele of the GSK-3β promoter region C are more susceptible to schizophrenia (*Tang et al., 2013*). In schizophrenia (SZ), carriers of the low-activity C allele variant have significantly higher brain volume in the temporal lobe, which is the brain parenchymal region with the most consistent morphological abnormalities in schizophrenia. The neuropathological process in this area develops rapidly at the onset of

the disease, suggesting that carrying the low-activity mutant C allele gene protects the brain from related neuropathological damage. GSK-3β is an important factor that affects the neuropathology of schizophrenia (*Benedetti et al., 2010*).

### GSK-3*β* and tumour therapy

GSK-3β is associated with the occurrence and development of tumours. Given its ability to negatively regulate oncogenic proteins, such as MYC and β-catenin, GSK-3β likely plays a suppressive role in tumourigenesis. Nevertheless, several investigations have demonstrated that GSK-3β may exhibit a positive regulatory role in tumourigenesis within human ovarian, liver, colon, and pancreatic carcinomas (*Luo, 2009*). Whether GSK-3β plays a tumour suppressor role or serves as a tumour promoter depends on the cell type and cellular context (*Takahashi-Yanaga, 2013*). Glioblastoma (GBM) is the most common primary intracranial tumour, originating from glial cells and accounting for approximately 81% of intracranial tumours (*Xu et al., 2020*). The expression levels of GSK-3β and Tyr216-phosphorylated GSK-3β are higher in GBM tissues than in nontumor brain tissues. GSK-3β in GBM often plays a pro-cancer role. Inhibiting the activity or expression of GSK-3β *in vitro* reduces the survival and proliferation of glioblastoma cells, as well as induces apoptosis of glioblastoma cells and delays the growth of mouse neuroblastoma tumours (*Dickey et al., 2011*). The phosphorylation of GSK-3β at Ser9 results in the inhibition of GSK-3β, which subsequently leads to the activation of glycogen synthase, thereby facilitating glycogen synthesis. Glioblastoma cells exhibit notably elevated levels of glycogen, and the accumulation of glycogen serves as a promoter for the growth of GBM (*Majewska & Szeliga, 2017*). In the clinical setting, it has been observed that the levels of GSK-3β and its phosphorylated form (Tyr216) are elevated in tumours compared to normal tissues. Significant inhibition of GSK-3β has been shown to reduce the survival and proliferation of glioblastoma cells, and this effect is accompanied by an upregulation of p53 and p21 expression (*Miyashita et al., 2009*). In addition, *Kotliarova et al. (2008)* reported that inhibition of GSK-3β activity results in c-MYC activation, leading to the induction of glioma cell death.

Of note, the role of GSK-3β in glioblastoma tumours is controversial (Fig. 3). *Li et al. (2010)* demonstrated that GSK-3β is highly expressed and activated during differentiation in sensitive C6 malignant glioma cells. Interference of GSK-3β activity with GSK-3β inhibitors or siRNA potently suppresses differentiation in sensitive C6 cells. Conversely, overexpression of a constitutively active form of GSK-3β (GSK-3β-S9A) mutant in differentiation-resistant U251 glioma cells restores their differentiation abilities (*Li et al., 2010*). *Zhao et al. (2015)* demonstrated that the levels of p-GSK-3β (Ser9) are significantly upregulated in glioma tissues compared to normal tissues. Ectopic expression of GSK-3β in glioma cells significantly inhibits tumour growth, which is accompanied by a decrease in β-catenin expression and downregulation of p-mTOR and p-p70S6K1 (*Zhao et al., 2015*). Consistently, inactivation or degradation of GSK-3β promotes glioblastoma invasion and epithelial-mesenchymal transition (EMT) in glioblastoma cells (*Li et al., 2019*; *Yang et al., 2019*). Furthermore, CHIR99021, a GSK-3β inhibitor, greatly increases glioma stem-like cell (GSLC) properties in patient-derived glioma samples (*Yang et al., 2020*). Nevertheless,

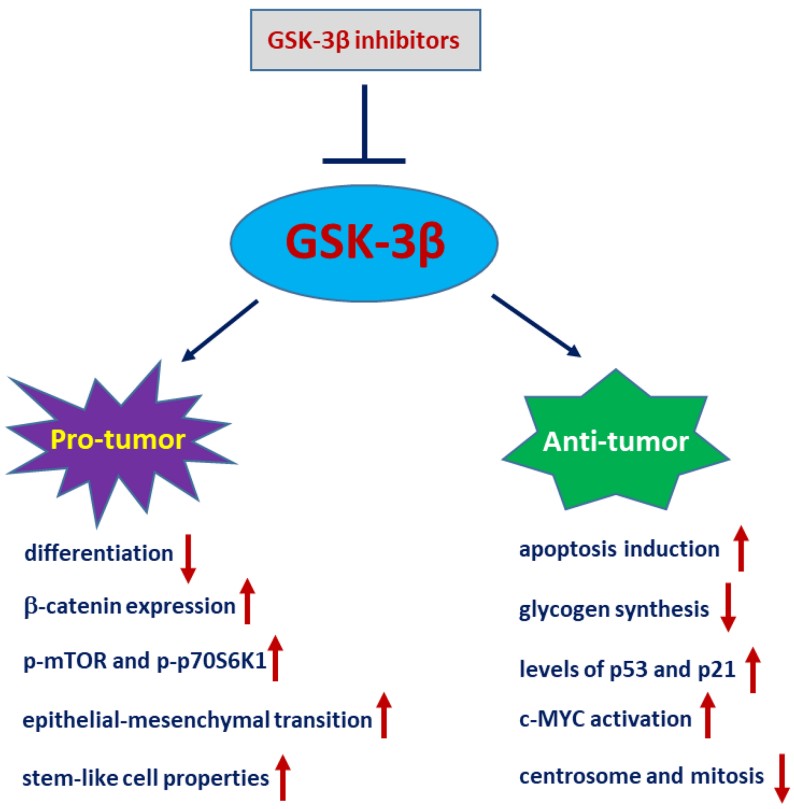

**Figure 3** GSK-3β plays a paradoxical role in glioblastoma.

increasing evidence also suggests that GSK-3β is an important molecule that leads to the malignant phenotype of GBM. AZD2858 is an effective adjunct to glioma radiotherapy at clinical doses. Most recently, studies have shown that GSK-3β inhibition by AZD2858 promotes glioma cell death by disrupting centrosome function and inducing mitotic defects (*Brüning-Richardson et al., 2021*).

## CONCLUSIONS AND PERSPECTIVES

In the present review, the possible underlying mechanisms that connect GSK-3β signalling with the physiopathology of the brain are discussed. From our perspective, the function of GSK-3β in brain pathophysiology is complicated and not entirely understood. GSK-3β is implicated in a variety of psychiatric and neurological disorders, emphasizing both its importance and complexity within the brain. It is meaningful to elucidate the basic biological mechanisms underlying GSK-3β in the brain because the increased understanding of the relationship between GSK-3β and the neural system greatly impacts the progress in treating several neurological diseases. Notably, there is a different opinion in the field regarding the effects of GSK-3β on cognition in the brain. For example, studies have shown that GSK-3β activity in the amygdala and hippocampus is needed for memory reconsolidation (*Hong et al., 2012*; *Xie et al., 2022*). With an increase in total GSK-3β and phosphorylated GSK-3β (Tyr216) in adult mice after exercise, *Zang et al. (2017)* observed

that there is a significant enhancement in adult neurogenesis and cognitive functions. In old mice, GSK-3β is needed for memory formation (*Koike et al., 2021*). These findings suggest that inhibition of GSK-3β in neurological disorders may bring potential risk of cognitive impairment. To some degree, these results also possibly explain the poor efficacy of GSK-3β inhibitors in preserving memory capacity in AD patients.

GSK-3β has emerged as a pivotal molecule with multifaceted roles in various cellular processes, extending from glycogen metabolism to critical signalling pathways, especially those involving β-catenin and NF-κB. The intricate regulatory mechanisms of GSK-3β, particularly through phosphorylation events, underscore its significance in both normal cellular homeostasis and pathological conditions. In the context of glioblastoma (GBM), the aberrant activation of the GSK3β/β-catenin pathway and the consequential modulation of key proteins, such as c-Myc and c-Jun, highlight the potential of GSK-3β as a therapeutic target. However, the role of GSK-3β in glioblastoma tumours is still controversial. Considering the carcinogenic risk of GSK-3β in gliomas, GSK-3β inhibitors should be used with caution as a treatment for glioblastoma. It is best to utilize an effective GSK-3β inhibitor that does not cause any carcinogenic risk while having an ideal anticancer effect through the appropriate dose range. Overall, the profound influence of GSK-3β on GBM malignancy, especially its implications in GSC survival, warrants further investigation to elucidate its complete role and pave the way for targeted therapeutic strategies in GBM management.

Comprehensive investigations are imperative to elucidate the precise signalling mechanisms regulated by GSK-3β and the mechanistic contributions of GSK-3β in the development of brain diseases. Urgent efforts are required to expedite advancements in substantiating the advantages of GSK-3β interventions and addressing the challenges associated with their implementation as a therapeutic strategy for brain diseases. Despite encouraging outcomes observed in preclinical studies utilizing GSK-3β inhibitors, caution must be exercised prior to their utilization, necessitating further examinations to assess both their beneficial effects and potential side effects during application. With the rapid development of technologies that allow elucidation of the characteristics and pattern of GSK-3β in brain diseases, the time has come to build on these findings to dissect the complexity of organisms and exploit the optimal therapy method.

While many studies have contributed to the understanding of how GSK-3β is involved in brain physiology and pathology, the recent development of conditional gene editing, molecular imaging, and optogenetic tools allows a more precise determination of the functional significance of these findings and application of them to improve the understanding and treatment of neurological disorders. Combination therapeutic strategies utilizing GSK-3β inhibitors hold great promise for the treatment of Alzheimer's disease (AD), Parkinson's disease (PD), and bipolar disorder (BD). The activity of GSK-3β is regulated by diverse stimuli, which can lead to its activation or inactivation. This phenomenon is intricately associated with the complex characteristics and genetic heterogeneity of brain disorders, particularly in the realm of brain tumours. Personalized genomics and diverse omics methodologies hold promise in identifying patients who may derive therapeutic benefits from GSK-3β inhibitor treatment. Ultimately, this knowledge

can guide clinical practitioners in selecting the most appropriate therapy for each individual patient.

### Funding

This work was supported by the Natural Science Foundation of China (No. 82204500), and the Key Discipline Construction Project of Pudong Health Bureau of Shanghai: Clinical Pharmacy (No. PWZxk2022-27). The funders had no role in study design, data collection and analysis, decision to publish, or preparation of the manuscript.

### Grant Disclosures

The following grant information was disclosed by the authors:
Natural Science Foundation of China: 82204500.
Key Discipline Construction Project of Pudong Health Bureau of Shanghai:
PWZxk2022-27.

### Competing Interests

The authors declare that they have no competing interests.

### Author Contributions

- Shenjin Lai conceived and designed the experiments, performed the experiments, analyzed the data, prepared figures and/or tables, authored or reviewed drafts of the article, and approved the final draft.
- Peng Wang analyzed the data, prepared figures and/or tables, and approved the final draft.
- Jingru Gong performed the experiments, analyzed the data, prepared figures and/or tables, and approved the final draft.
- Shuaishuai Zhang conceived and designed the experiments, authored or reviewed drafts of the article, and approved the final draft.

### Data Availability

This is a literature review.

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
