# Peer review of "New insights into the role of GSK-3β in the brain: from neurodegenerative disease to tumorigenesis"

_PeerJ, doi:10.7717/peerj.16635_

## Round 0.1 · original submission · Major Revisions

All the reviewers comments must be carefully addressed as recommended.

**Language Note:** The review process has identified that the English language must be improved. PeerJ can provide language editing services - please contact us at copyediting@peerj.com for pricing (be sure to provide your manuscript number and title). Alternatively, you should make your own arrangements to improve the language quality and provide details in your response letter. – PeerJ Staff

Reviewer 1 ·

Basic reporting

I am thanking you to give me a chance to read and review this interesting article, the article is about a very interesting subject and I do not have any conflict of interest with the authors. Besides, I have not found any significant similarity index in the manuscript. However, there are some points and few suggestions which should be considered:
1- Abstract has been written very well, but keywords should be chosen and written from the words which are not mentioned in the Title.
2- Keywords should be written on the basis of alphabetic order.
3- Paragraphing should be done according to the new point. When authors have started to write about a new point and findings, they should start a new paragraph. Also, paragraphs should not be started with new reference.
4- The main text is interesting, and it has written very well, however, there are some few mistakes and grammatical errors which should be considered and corrected, and I think authors should double-check the manuscript to reduce these few grammatical errors.
5- References in the text should be written on the basis of order, in this manuscript, authors have jumped from Reference 115 to 143, which should be corrected.
6- Some of the references are not on the basis of journal s format. And, it is suggested that authors use and write DOI for all articles.

Experimental design

It is interesting and appropriate.

Validity of the findings

It is acceptable and it is OK.

Additional comments

Please, consider the points that I have mentioned, after correcting and considering these points, the manuscript can be suggested and recommended for publication.

Reviewer 2 ·

Basic reporting

The article has written very well. But its Abstract and Conclusion Must be revised, they are very similar to each others, and Conclusion is not enough, and it should be revised completely, what is the recommendations of authors for future researches? What is the suggestion of authors to improve the researches in future?

I also recommend authors double-check the English of the manuscript, there are some mistakes in the text, however, the article has just need MINOR REVISION.
The article s topic is very important and interesting.

Experimental design

Acceptable

Validity of the findings

Acceptable and interesting

Additional comments

I also recommend authors double-check the English of the manuscript, there are some mistakes in the text, however, the article has just need MINOR REVISION.
The article s topic is very important and interesting.

Reviewer 3 ·

Basic reporting

This paper addresses an intriguing topic related to GSK-3β in neuronal physiological process, and in CNS diseases. While this review appears timely and comprehensive, it could benefit from providing more specific details about the mechanisms involved. Additionally, discussing more about potential clinical applications would enhance the paper's significance.

Experimental design

No comment

Validity of the findings

1. In general, the article covers many aspects, including GSK-3β in synaptic plasticity and memory formation, neuroinflammation,neuronal survival and neurogenesis,cerebral ischemia, and tumorigenesis. But some sections are less detailed and not sufficiently in-depth, such as section 3.1 and 3.3. It is recommended to provide a more thorough description of the relevant mechanisms and clinical applications.
2. In fact, most of the content in Section 3.6 and Section 4.5 (which consists of only one paragraph) summarizes GSK-3β and glioblastoma. What is the relationship between these two sections, and can they be integrated into a comprehensive review that bridges the mechanism to clinical applications?
3. In section 4.1, while the paper discusses the involvement of GSK-3β in tau hyperphosphorylation and amyloid production, it does not go into detail about the molecular mechanisms underlying these processes. Providing some insights into the specific pathways or interactions involved would be informative.
4. Section 4.2 is lack of mechanistic description and potential clinical application targets.
5. What is the AKT- GSK-3β signaling pathway functions in schizophrenia? Offering a brief explanation of the pathway's role in synaptic plasticity and neuronal function would enhance understanding.

Additional comments

1. Pls ensure that abbreviations are only used when the word appears at least a few times.
2. Figure 2 is too simplistic; perhaps disease models could be added to provide a summarized overview.

Reviewer 4 ·

Basic reporting

The authors provide a comprehensive and detailed insight into the multifaceted roles and regulatory mechanisms of GSK-3β in various neurological disorders. The inclusion of extensive references and pertinent illustrations emphasizes the breadth and depth of research in this domain. The figure showcasing regulatory mechanisms of GSK-3β offers an intelligible portrayal of the molecular intricacies involved. Furthermore, the table highlighting representative GSK-3β inhibitors is a valuable resource, delineating their potential therapeutic applications across multiple brain diseases. The authors have systematically covered the vast literature, making it a valuable resource for both novices and experts in the field. However, a few aspects and suggestions might enhance the comprehensiveness and clarity of the manuscript.
While this paper offers a broad overview of GSK3β's role, its extensive scope sacrifices depth, making it challenging to discern the central thesis. To enhance the clarity and focus of the article, it would be advisable to trim content that isn't central to the main argument or objective. Additionally, the inclusion of illustrative diagrams or schematic representations could significantly aid readers in grasping complex concepts and the overarching narrative.

Experimental design

No comments.

Validity of the findings

Glycogen synthase kinase-3 (GSK-3) plays a vital role in various cellular processes, tightly regulated by complex signaling pathways. Dysregulation of GSK-3 is linked to diseases like Alzheimer's and other neurodegenerative conditions. GSK-3 acts as a central hub, integrating multiple signaling systems. Inhibiting GSK-3 is seen as a promising treatment strategy for neurodegenerative diseases.
1. While the article provides a comprehensive overview of the regulatory mechanisms of GSK-3β, the depth of discussion on each mechanism seems to be lacking. It would benefit readers to delve deeper into key areas for a more thorough understanding. In the section discussing GSK-3β's subcellular localization, the article states that GSK-3β exhibits heightened activity within the nucleus and mitochondria. While this is an interesting point, the article could benefit from more elaboration on how the localization specifically affects GSK-3β's kinase activity. Are there particular substrates or interacting proteins present in these subcellular locations that contribute to this heightened activity? This aspect requires further discussion and clarification.
2. In the sections "2. Regulation Mechanisms of GSK-3β Action" and especially "3.6 GSK-3β and tumorigenesis", it would be highly beneficial to include a schematic diagram illustrating the discussed mechanisms and pathways. An illustrative representation would not only enhance reader comprehension but would also allow for a focused discussion on specific pathways, providing depth and clarity. Instead of a generalized discussion, selecting a few key pathways for in-depth elaboration would strengthen the article's coherence and impact.
3. GSK-3β emerges as a pivotal molecule with multifaceted roles in various cellular processes, extending from glycogen metabolism to critical signaling pathways, notably those involving β-catenin and NF-κB. Its intricate regulatory mechanisms, particularly through phosphorylation events, underscore its significance in both normal cellular homeostasis and pathological conditions. In the context of glioblastoma (GBM), the aberrant activation of the GSK3β/β-catenin pathway and the consequential modulation of key proteins, such as c-Myc and c-jun, highlight its potential as a therapeutic target. The protein's profound influence on GBM malignancy, especially its implications in GSCs survival, warrants further investigation to elucidate its complete role and pave the way for targeted therapeutic strategies in GBM management. Please discuss.
4. Given that the central theme of this manuscript revolves around the relationship of GSK-3β in tumorigenesis, it would be pertinent for the section "4. GSK-3ò and its Inhibitors in Brain Disease" to emphasize research advancements specifically related to tumors. This focus would not only enhance the overall coherence of the article but also provide readers with a more in-depth understanding of the potential therapeutic implications of GSK-3β inhibitors in tumor-related brain diseases.
5. The therapeutic potential of enzastaurin (LY317615), a GSK3β and PKC-β inhibitor, has been rigorously tested. While initial clinical trials showcased its promise and tolerability, subsequent phase III trials were more tepid in their outcomes, underscoring its limited efficacy as a monotherapy. As a result, the scientific community has pursued combination therapies involving enzastaurin, although clear benefits for patients remain elusive. The findings suggest that while GSK3β holds promise as a therapeutic target, optimal strategies for its inhibition in GBM treatment still warrant further exploration. Please discuss.
6. While this paper offers a broad overview of GSK3β's role, its extensive scope sacrifices depth, making it challenging to discern the central thesis. To enhance the clarity and focus of the article, it would be advisable to trim content that isn't central to the main argument or objective. Additionally, the inclusion of illustrative diagrams or schematic representations could significantly aid readers in grasping complex concepts and the overarching narrative.

---

## Round 0.2 · accepted · Accept

The manuscript is now suitable for publication.

Reviewer 1 ·

Basic reporting

The article has been written and revised very well.

Experimental design

It is OK.

Validity of the findings

It is OK, and it is very well-written.

Additional comments

The article is OK, and it can accepted and publish in present format.

Reviewer 2 ·

Basic reporting

The article has written very well, the quality of figures and tables are acceptable, and I have not found high significant similarity index in the manuscript. Both parts of Abstract and Conclusions have clearly illustrated the manuscript. References are OK, and authors have used enough references and references are updated.
There are just few grammatical error in some sentences, and also should double check the manuscript again or ask an English-speaking editor to edit the manuscript for them.
The article can be accepted.

Experimental design

It is well-written and OK.

Validity of the findings

The information and findings in the manuscript are new and very informative, and the article is very interesting and informative for readers.

Additional comments

Authors just need to check English language of the manuscript. Some sentences need revisions and some english words in the manuscript have not written very well.

Reviewer 3 ·

Basic reporting

1. Basic reporting
Upon reviewing the revised Manuscript Ref. No.: #89459 titled "New insights into the role of GSK-3β in brain: From neurodegenerative disease to tumorigenesis," I find the revisions and the authors' responses to be thorough, with the manuscript now meeting the high standards of clarity and context required. The use of clear, professional English and the structured presentation of the content, including figures, tables, and raw data, are commendable. The review is relevant to the journal's scope, reflecting broad and cross-disciplinary interest, and it provides a fresh perspective on the field, which has not been reviewed recently. The introduction is well-crafted, clearly defining the subject and its significance to the intended audience. These enhancements to the manuscript substantiate its value to the field, and I endorse its acceptance for publication.

Experimental design

The manuscript showcases a robust study design, aligning perfectly with the journal's aims, and is marked by a thorough investigation that meets high ethical and technical standards. Detailed methodological descriptions ensure reproducibility, while the logical organization of the content confirms a systematic approach to research.

Validity of the findings

The findings of the manuscript titled "New insights into the role of GSK-3β in brain: From neurodegenerative disease to tumorigenesis" are presented with a clear connection to the stated research question, indicating a logical flow from hypothesis to conclusion. The study contributes to the existing body of literature with its replication efforts and clarifies the rationale behind its approach. The argumentation within the paper is coherent, addressing the objectives set in the Introduction. In the Conclusion, the manuscript appropriately points out remaining questions and potential areas for future investigation, which could serve to further substantiate its findings.

Reviewer 4 ·

Basic reporting

I have reviewed the revisions and responses provided by the authors for Manuscript Ref. No.: #89459 titled "New insights into the role of GSK-3β in brain: From neurodegenerative disease to tumorigenesis." The authors have satisfactorily addressed all the concerns and queries raised in my previous review. The corrections and language improvements, as indicated in red and blue respectively in the marked manuscript, are well-executed and enhance the clarity and quality of the paper.

I am particularly impressed with the thoroughness of the revisions and the attention to detail in addressing the specific points raised. The additional information and clarifications provided have significantly improved the manuscript and its contribution to the field.

Given these substantial improvements and the quality of the work presented, I am pleased to recommend the acceptance of this manuscript for publication.

Experimental design

The study design is commendable for its adherence to the journal's scope, rigorous and ethical investigation, and clear methods enabling replication. Its comprehensive survey methodology, proper citation of sources, and logically structured review reflect a meticulous approach, underpinning the study's reliability and contribution to the field.

Validity of the findings

The manuscript provides a coherent presentation of findings related to GSK-3β's role in the brain, drawing a direct line from the research questions posed to the evidence provided. It contributes constructively to the field through replication studies and justifies its approach within the broader scientific context. The narrative is consistent, meeting the aims outlined in the introduction, and the conclusions are substantiated by the results. Unanswered questions and future research opportunities are duly noted, suggesting a path forward for subsequent inquiry.